# Editable Temporal Graph Neural Network for Interactive Time Series Forecasting

## Abstract

Time-series data often contain multiple dependent variables that evolve with persistent relationships. Modeling such relational information with graphs can improve the forecasting accuracy significantly, while providing explainable insights to the human users on the underlying dependencies. Conventional graph temporal models propose learning such relational information from the data, which might not capture the correct relations. Ideally, it is desirable to provide mechanisms for users to modify the learned relationships before using the models for forecasting, especially in the high-stakes decision making scenarios. In this paper, we propose a novel model, Editable Temporal Graph Neural Network (ETGNN), and an editing algorithm that allows users to make edits on the learned graphs before obtaining forecasts, in a superior way compared to the alternatives. A key novelty of ETGNN model is the use of global embeddings in both graph learning and time series forecasting, which allows fast editing the forecasts in a local region without affecting unrelated forecasts. We show that after editing, ETGNN can achieve close-to-perfect accuracy for latent graph prediction (with the F1-score of larger than 0.9) and reduce the time series forecasting error by 10%~40% over multiple datasets, with much lower editing cost compared to retraining. Overall, ETGNN provides a convenient mechanism to incorporate user feedback to improve the accuracy and explainablity of complex forecasting tasks.

## 1 Introduction

Most time-series datasets contain multiple covariates evolving based on certain relationships. For example, increase in the traffic flow at one road intersection might signal the upcoming increase in the neighboring road intersections, or increase in the sales of new laptops or phones might yield strong signals on the sales of corresponding power sources. Utilizing the relational information in time-series modeling can significantly improve the accuracy of time series forecasting (Zügner et al., 2021). Hence, most prior work is based on forecasting time series of multiple entities through graph-temporal modeling, i.e., incorporating graph structure information into time series forecasting models (Li et al., 2018; Yu et al., 2018). Nonetheless, these methods heavily rely on known or pre-computed graphs, which are not available in many practical scenarios. To overcome this issue, recent studies proposed learning the latent relational graph and making forecasts simultaneously (Wu et al., 2020; Shang et al., 2020; Satorras et al., 2022). The learned graphs reveal the complex underlying relationships between the time series entities. As opposed to the methods that use known graphs, the learned graphs can provide unique explainability capabilities to the human users in a succinct way, providing easy-to-understand relational information. This might be used to build trust and better verify the model behavior (Kipf et al., 2018).

As the latent relational graphs are not learned by direct supervision, but learned implicitly with the goal of improving the forecasting accuracy, they can be inaccurate in capturing the actual relationships. Especially if the datasets are small or non-stationary, such graph-temporal models might be prone to overfitting to spurious relationships in the training data, thereby achieving poor generalization. In many real-world scenarios, it is desirable to correct the wrong relationships learned by such models, to improve the accuracy and explainablity of the model using users feedback. For example, retail data analysts might have rich domain knowledge on whether the sales of certain products should have strong relations or not, and might want to

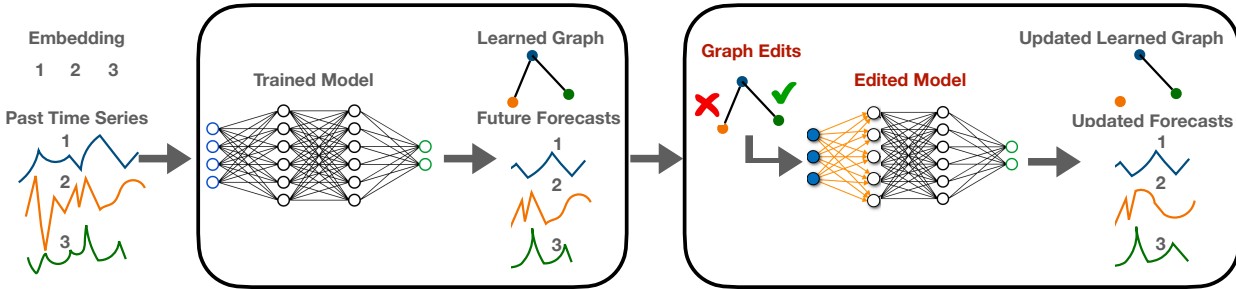

Figure 1: Overview of the proposed time series model editing framework. We propose a novel model, named editable temporal graph neural network (ETGNN), and a fast model editing algorithm. Given global embeddings and past time series, we first train the ETGNN model to jointly forecast future time series and predict a latent relational graph. With user feedback on the latent relational graph, the model can be edited to faithfully reflect the true relational information and correspondingly update the future time series forecasts.

correct the mistakenly inferred relations from the model, and speculate the changes in the predicted time series. The capability of having an "editable" graph temporal model would be of significant value to such human-in-the-loop applications where injecting expert knowledge into the model is needed. Such a model should be able to benefit from the direct supervision on the graph accuracy in a superior way and ideally its cost should be much lower than standard model training or fine-tuning so that frequent updates can be done by the users for more effective human-in-the-loop decision making.

To incorporate direct supervision from users on particular inputs with low computational cost, model editing (Mitchell et al., 2022; De Cao et al., 2021) was introduced for text data. Their primary goal is to incrementally update a trained model, revise its prediction to reflect the factual knowledge from the new edit data. On the other hand, our model editing setting is completely different – our edit data (extra supervision) are on the latent relational graph rather than the time series itself. This is mainly due to the characteristics of multivariate time series. Different from discrete text data, time series is continuous-valued. Hence direct edits on the time series is both difficult to control and hard to interpret. The relational graph, on the other hand, is discrete and much easier to interpret. For example, we can simply add or remove the edge representing the dependency between the time series. Edits on the relational graph would also change the forecasts indirectly through updated graph attention rather than directly on the observations.

In this paper, we propose a temporal graph model editing framework, as shown in Figure 1. Our framework first trains a graph temporal model to forecast time series and infer a relational graph, and then edits the model with gradient descent based algorithm from the user feedback on the true relational information. We propose a novel deep sequence model, named *Editable Temporal Graph Neural Network* (ETGNN) that can perform joint graph learning and time series forecasting. Most importantly, ETGNN can efficiently incorporate extra supervisions on the learned graph and further improve the forecasting performance. Specifically, our model consists of two inter-connected components: graph learning module and time series forecasting module. The graph learning module computes a global embedding vector for each entity, and infers a latent relational graph indicating the latent relations between every pairwise entities. The global embeddings and the inferred graph are then fed into the time series forecasting module to forecast the future of multivariate time series. The use of global embeddings in both graph inference and time series forecasting modules facilitates editing the two modules based on the user feedback. We demonstrate the success of our proposed framework and model in efficiently editing predictions on multiple real-world time-series datasets. Our contributions in this work can be summarized as:

- Introducing the novel problem setting of model editing for graph-based time series forecasting – we focus on joint graph learning and time series forecasting that also allows for extra supervision on the learned graph.

- Proposing an efficient model editing framework – we introduce a novel editable temporal graph neural network that is well-suitable for fast model editing and an editing algorithm based on minimizing the combination of minimum squared error loss on time series and cross entropy loss on graphs with gradient descent.

- Conducting extensive experiments to show the effectiveness of the proposed method – through experiments on one physics simulation dataset and two traffic datasets, we show that after editing all mistakenly inferred latent relations, ETGNN model achieves close-to-perfect accurate latent graph prediction (with the F1-score of larger than 0.9) and reduce the time series forecasting error by 10%∼40% after being edited. Also, compared to existing temporal graph neural network baselines, our model shows greater benefits in improving the time series forecasting with editing.

## 2 Related Work

**Multivariate Time Series Forecasting.** Exploiting relational information in multivariate time series (MTS) has been studied extensively for spatiotemporal forecasting, especially for traffic forecasting scenario (Li et al., 2018; Yu et al., 2018) , at which different sensors are represented as nodes in the graph with the edges corresponding to the spatial proximity between sensor locations. In this scenario, the graph is known or pre-computed using the road-network information. Wu et al. (2020); Shang et al. (2020); Satorras et al. (2022) further consider the scenario where the graph is unknown but can be learned from data. They propose to combine relational inference techniques (e.g. Neural Relational Inference (Kipf et al., 2018)) with time series forecasting. However, a recent study on joint graph learning and forecasting by Zügner et al. (2021) found that the learned graph might not align with the truth graph. The impact of inferring graphs together with forecasting time series also varies significantly among datasets. These observations motivate for "model editing": modify the learned graph with the extra supervision from the user domain knowledge and improve the forecasts accordingly.

**Temporal Graph Neural Networks.** Graph neural networks (GNNs) seek to learn representations over relational data (see several recent surveys on GNNs and the references therein, e.g., Bronstein et al. (2017); Zhang et al. (2018); Wu et al. (2019); Goyal & Ferrara (2018)). The majority of the GNNs research is on static graphs instead of temporal graphs. Even for temporal graph neural networks such as DyRep (Trivedi et al., 2018) and TGAT (Xu et al., 2020), the tasks are limited to classic graph analysis such as node/edge classification rather than time series forecasting. Further-

Table 1: A conceptual comparison of different temporal graph neural networks.

| Model | Forecast series? | Learn graph? | Editable? |
|-------|------------------|--------------|-----------|
| TGAT  | ✗ | ✗ | ✗ |
| NRI   | ✗ | ✓ | ✗ |
| MPGNN | ✓ | ✓ | ✗ |
| GTS   | ✓ | ✓ | ✗ |
| ETGNN | ✓ | ✓ | ✓ |

more, most existing works in GNNs assume the graph is *known* and train the model in a supervised fashion. In contrast, relational inference aims to discover the latent relations and is therefore unsupervised. The seminal work of Neural Relational Inference (NRI) (Kipf et al., 2018) uses neural networks to reason in dynamic physical systems. Alet et al. (2019) reformulate NRI as meta-learning and propose simulated annealing to search for graph structures. Relational inference is also posed as Granger causal inference for sequences (Louizos et al., 2017; Löwe et al., 2020). Motivated from NRI, some other works (e.g., MTGNN (Wu et al., 2020), GTS (Shang et al., 2020)) propose temporal graph neural networks that can jointly infer the relational graph structure and forecast time series from the inferred graph. Nevertheless, different from our proposed ETGNN, none of the aforementioned methods can take in extra supervision and allow *editable* temporal graph learning. See Table 1 for a conceptual comparison of different temporal graph neural networks.

**Model Editing / Adaptation.** The term "model editing" originates from the natural language processing (NLP) applications where it is proposed to correct the errors made by large language models. This capability allows for scalable adjustment of model behavior after deployment. The edit dataset is provided as a small set of new examples with the correct labels. Having observed that fine-tuning on the new examples can lead

to over-fitting, Sinitsin et al. (2019) propose editable neural network with a gradient-based editor, where the parameter updates use additional constraints for reliability and locality. Mitchell et al. (2022) propose a meta-learning based method for model editing. They learn an *editor* network to transform fine-tuning gradients to parameter updates. Sotoudeh & Thakur (2021) investigate the provable repairing of a deep network, which repairs a network to construct a new network that satisfies given specifications. Our work shares similar motivations for model editing, but has a very different setting. Our edit data is not given as a set of labeled data (time series in our case), but the latent graph learned from data.

## 3 Preliminaries

### 3.1 Forecasting with Learned Graphs

In this work, our problem of interest is multivariate time series forecasting. In this problem, we aim to forecast the future time series of multiple entities from their past time series. Specifically, suppose we have $N$ entities and the set of complete time series sequences are $\{X_1, X_2, \cdots, X_N\}$, each of which is a stacking of $I$-dimensional feature vectors along time. Our target is to learn a time series forecasting model so that given the time series $X_{\text{in}} = \{X_i^{t-L+1:t}\}_{i=1}^N$ from the time step $t - L + 1$ to $t$, where $X_i^{t-L+1:t} \in \mathbb{R}^{L \times I}$, the model can predict the future $H$-step time series $X_{\text{out}} = \{X_i^{t+1:t+H}\}_{i=1}^N$, where $X_i^{t+1:t+H} \in \mathbb{R}^{H \times I}$. To model the dependencies among the entities, we can represent their latent relations as a graph whose nodes correspond to entities and edges are represented by the adjacency matrix $A \in \{0, 1\}^{N \times N}$. Here, $A_{ij} = 1$ indicates that there exist dependencies from the $i$-th to the $j$-th entity, or how the time series $X_j$ proceeds will be affected by $X_i$. $A_{ij} = 0$ indicates that such dependencies do not exist between them. Leveraging the dependency graph $A$, we can improve the performance of forecasting $X_{\text{out}}$ from $X_{\text{in}}$.

However, in real-world applications, the dependency graph is not always available. Therefore, many works propose to jointly learn the latent relational graph and forecasting multivariate time series (Wu et al., 2020; Shang et al., 2020; Satorras et al., 2022). Specifically, we introduce a graph inference model $q_\phi$ with parameters $\phi$ to infer the adjacency matrix $A$ of the latent relational graph. With the inferred graph $A$, time series forecasting is done by a time series forecasting model $p_\theta$ with parameters $\theta$. The problem of forecasting with learned latent graphs aims to optimize both models $q_\phi$ and $p_\theta$ simultaneously so that $p_\theta$ can accurately forecast future time series from the past with the graph learned by $q_\phi$. Here we assume the dependencies among entities are fixed across time steps and the latent graph $A$ is a global graph structure.

### 3.2 Model Editing

Generally, model editing refers to updating the parameters of a model so that its prediction outputs for some input data samples can be altered. Formally, we have a trained neural network model $f_\theta$ with parameters $\theta$, and a dataset $D_e = \{(X_i, Y_i)\}_{i=1}^M$ with $M$ data-label pairs. Usually, $D_e$ is the collection of data samples whose labels are mistakenly predicted by $f_\theta$. Model editing requires us to update the model parameters $\theta$ to $\theta'$ so that $f_{\theta'}$ makes correct predictions on $D_e$, i.e., for any $(X, Y) \in D_e$, $f_{\theta'}(X) = Y$, while the prediction for data not belonging to $D_e$ is not changed. An application scenario of model editing is correcting factual knowledge in large language models (De Cao et al., 2021; Mitchell et al., 2022).

In this work, however, we are not studying editing the direct output time series of the temporal graph neural network model. Different from text data in NLP applications, providing editing data of time series is much more difficult. On the other hand, editing the inferred latent relational graph based on domain knowledge is much easier. Hence, we will formulate and focus on a new model editing problem of correcting the latent relational graph inferred by a temporal graph neural network model.

## 4 Editable Temporal Graph Neural Network

In this section, we propose a framework to edit temporal graph neural network models in multivariate time series forecasting. We will first formally describe the problem of model editing on latent relational graphs. Next, we introduce a novel model, named editable temporal graph neural network (ETGNN), and an efficient

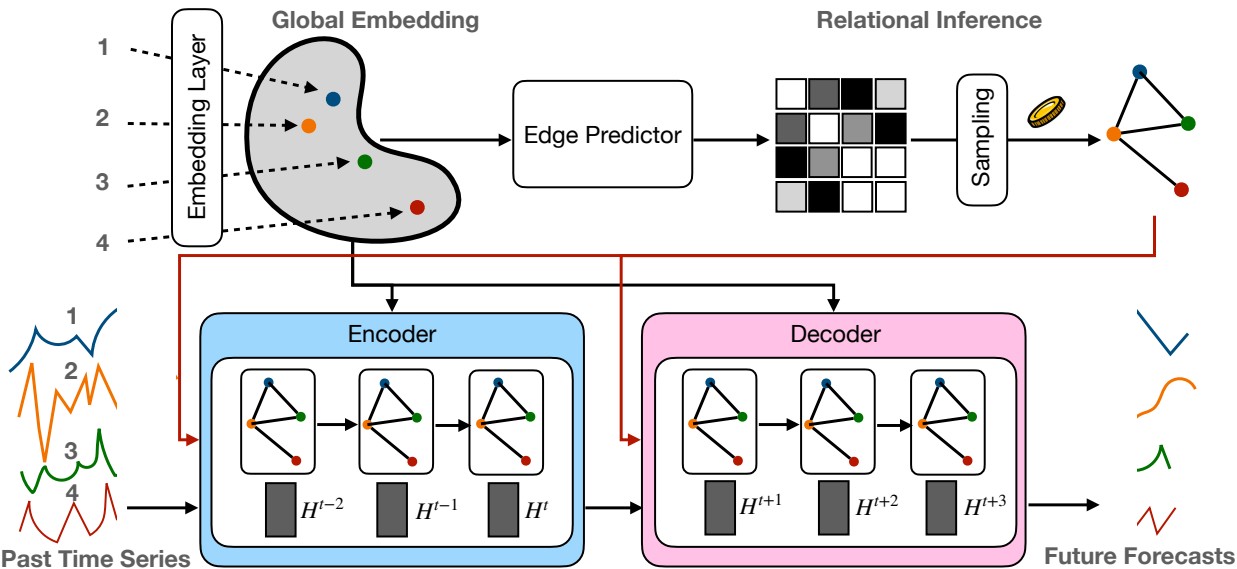

Figure 2: An illustration of the proposed ETGNN model.

algorithm for fast model editing. To our best knowledge, we are the first to propose a practical solution to the model editing problem in multivariate time series forecasting.

## 4.1 Model Editing on Latent Graph

In this work, we focus on editing the multivariate time series forecasting model to alter its *learned* latent relational graph, where the model is trained by joint graph learning and time series forecasting, as described in Section 3.1. This problem requires updating the model parameters using additional supervision on the latent graph. Let's suppose we are given a temporal graph neural network model, which is composed of the graph inference module $q_\phi$ with parameters $\phi$ and the time series forecasting module $p_\theta$ with parameters $\theta$. The model has already been trained on a time-series dataset without supervision on the latent relational graph. Given the adjacency matrix of the latent relational graph inferred by the trained model, $A$, we consider users editing some elements in $A$ with their domain knowledge. Note that in practice, users may only edit the latent relations of a few entities in a local region of the graph. Let the edited adjacency matrix be $A'$, our target is to update the parameters $\phi, \theta$ of the model to new parameters $\phi', \theta'$ so that the latent relational graph inferred by $q_{\phi'}$ is $A'$ and the forecasting performance of $p_{\theta'}$ is better than $p_\theta$. When updating models, we only allow changes on the edited elements of $A$ while keeping the rest intact. Hence, our model parameter updates have the following principles:

- For any $1 \le i, j \le N, i \ne j$, if $A_{ij} \ne A'_{ij}$, i.e., the latent relation from the $i$-th to the $j$-th entity needs edit, we update $q_\phi$ to changes its prediction from $A_{ij}$ to $A'_{ij}$.

- For any $1 \le i, j \le N, i \ne j$, if $A_{ij} = A'_{ij}$, i.e., there is no edit on the latent relation from the $i$-th to the $j$-th entity, the new model $q_{\phi'}$ should preserve the old prediction $A_{ij}$.

We solve this model editing problem by a novel editing-suited model architecture, and an efficient gradient descent based editing algorithm, presented in Section 4.2 and 4.3 , respectively.

## 4.2 Temporal Graph Neural Network

We propose ETGNN, a novel deep sequential model that is particularly well-suited for the model editing problem discussed in Section 4.1. As shown in Figure 2, our proposed model consists of two inter-connected

components: a latent relational graph inference module and a time series forecasting module. Next, we explain the details of the model architecture and highlight the features that facilitate model editing.

**Global Embeddings.** In our proposed ETGNN model, we assign a unique $Q$-dimensional embedding vector to each of the $N$ entities, denoted by $\{g_i\}_{i=1}^{N}$. All these global embedding vectors are trainable parameters of the model, and they directly affect the calculation in the latent relational graph inference module and time series forecasting module. Particularly, in our model, the embedding vector $g_i$ of the $i$-th entity has large impact on the prediction of the latent relations of the $i$-th entity in the graph, and the forecasts of the time series of the $i$-th entity.

Such designs of making global embedding vectors only have significant impact on the prediction (including the predictions of both latent relations and time series) of their corresponding entities facilitates model editing. First, as described in Section 4.1, users may only edit the latent relations of a few entities but keep all other latent relations unchanged. In this case, we expect that updating of model parameters would lead to significant changes in the predictions of those involved entities, but the changes of the predictions for other entities are small. Intuitively, this becomes easier if through modifying the global embedding vector of any entity, we can achieve making large changes to its predictions but minor changes to the predictions of the other entities. In other words, the use of global embeddings helps controlling the graph region that will be affected by the editing. Second, using global embedding vectors in both latent graph inference and time series forecasting modules facilitates the model to quickly adjust time series forecasts. In our problem setting, the user does not edit output time series but the latent relational graph, as the ultimate target is to reduce the time series forecasting error by offering an improved latent graph. Particularly, it is expected that for the entities with high time series forecasting error, their latent relations are more likely to be mistakenly predicted so users will edit their latent relations. For these entities, because their global embeddings affect their predictions in two modules, we can make large impacts to the predicted latent relations and time series simultaneously by making more modifications to their global embeddings. We show that sharing global embeddings in both modules leads to larger reduction in time series forecasting error in Section 5.3.

**Latent Graph Inference.** In the latent relational graph inference module, the adjacency matrix $A$ of the latent relational graph is inferred from a graph edge prediction model which takes as input the concatenation of two global embedding vectors. Specifically, to predict the element $A_{ij}$ of the adjacency matrix $A$, we first concatenate the embedding vectors $g_i$ and $g_j$, then feed them into the graph edge predictor model, which is a multi-layer perceptron (MLP) network. The direct output from this model is a probability value $p_{ij} \in (0, 1)$, and $A_{ij}$ are sampled from the Bernoulli distribution with the parameter $p_{ij}$. In this way, we can predict the latent relationships between all pairwise entities among all $N$ entities and obtain the complete latent relational graph $A$. Clearly, the computation process of predicting $A$ ensures that the embedding vector $g_i$ can only affect the prediction of the latent relations of the $i$-th instance.

To allow the gradient pass through the discrete sampling process from $p_{ij}$ to $A_{ij}$, we apply the Gumbel-Softmax (Jang et al., 2016) trick. Formally, $A_{ij}$ is sampled as

$$\hat{p}_{ij} = \frac{1}{1 + \exp\left(-\left(\log p_{ij} + G\right)/\tau\right)}, \quad A_{ij} = \left\lfloor \hat{p}_{ij} + \frac{1}{2} \right\rfloor, \tag{1}$$

where the random variable $G \sim \mathrm{Gumbel}(0,1)$ is sampled from the standard Gumbel distribution, $\tau$ is a temperature hyperparameter and $\lfloor \cdot \rfloor$ is floor function, i.e., $\lfloor x \rfloor$ maps $x$ to the largest integer that is not larger than $x$. The sampling in Eq. (1) is equivalent to directly sampling from Bernoulli distribution, and the gradients w.r.t. $\hat{p}_{ij}$ is used to approximate the gradients w.r.t. $A_{ij}$.

**Graph Recurrent Layer.** In the time series forecasting module, we use a sequence-to-sequence framework with Gated Recurrent Units (GRU) (Cho et al., 2014) for time series forecasting. Particularly, we replace the matrix multiplications of standard GRU with graph attention (GAT) network (Veličković et al., 2018; Brody et al., 2022). Specifically, at the time step $t$, let the input time series of $N$ entities be $X^t \in \mathbb{R}^{N \times I}$, and the hidden state from the time step $t-1$ be $H^{t-1} \in \mathbb{R}^{N \times P}$, the GRU model updates $H^{t-1}$ to new hidden

states $H^t$ by running GAT network on the latent relational graph $A$ as

$$r^t = \sigma\left(\text{GAT}_r\left(\left[X^t, H^{t-1}\right], A, \{g_i\}_{i=1}^N\right) + b_r\right), \quad u^t = \sigma\left(\text{GAT}_u\left(\left[X^t, H^{t-1}\right], A, \{g_i\}_{i=1}^N\right) + b_u\right),$$
$$C^t = \tanh\left(\text{GAT}_C\left(\left[X^t, r^t \odot H^{t-1}\right], A, \{g_i\}_{i=1}^N\right) + b_c\right), \quad H^t = u^t \odot H^{t-1} + (1 - u^t) \odot C^t. \tag{2}$$

Here, $\sigma$ is the sigmoid function, $[\cdot, \cdot]$ is concatenation operation, $r^t, u^t, C^t$ are reset gate, update gate, and the candidate vector at time $t$, $b_r, b_u, b_c$ are bias vectors, $\text{GAT}_r, \text{GAT}_u, \text{GAT}_C$ are three GAT networks used to compute reset gates, update gates, and candidate vectors for all time steps. Note that the three GAT networks take the global embedding vectors $\{g_i\}_{i=1}^N$ of $N$ entities as one of their inputs.

**Graph Attention Network.** We adopt the GATv2 architecture proposed by Brody et al. (2022) for the three GAT networks in Equation (2). Generally, the GAT network takes a graph with initialized node features as inputs, and computes node embeddings with $L$ attention-based message passing layers. In ETGNN, the GAT network is used as the basic unit of GRU based time series forecasting module. Formally, at time $t$, for node $i$ (the $i$-th entity) in the graph, the computation process of the $\ell$-th ($1 \leq \ell \leq L$) message passing layer can be described as:

$$\text{attn}_\ell(i, j) = g_i^T \text{LeakyReLU}\left(W_\ell E_{\ell-1,i}^t + V_\ell E_{\ell-1,j}^t\right), \quad \alpha_{ij} = \frac{\exp\left(\text{attn}_\ell(i, j)\right)}{\sum_{j \in \mathcal{N}(i)} \exp\left(\text{attn}_\ell(i, j)\right)},$$
$$E_{\ell,i}^t = W_\ell E_{\ell-1,i}^t + \sum_{j \in \mathcal{N}(i)} \alpha_{ij} \cdot V_\ell E_{\ell-1,j}^t. \tag{3}$$

Here, $\mathcal{N}(i) = \{j | j \neq i, 1 \leq j \leq N, A_{ij} = 1\}$, $W_\ell, V_\ell$ are trainable parameter matrices, $E_{\ell-1,i}^t \in \mathbb{R}^Q$ is the node embedding vector of node $i$ outputted from the $(\ell - 1)$-th message passing layer. Note that in Equation (3), we use the global embedding vector $g_i$ of the $i$-th entity as the learnable query vector of node $i$ to compute attention scores. Hence, $g_i$ have a large impact on the final node embeddings of node $i$ and the forecasted time series of the $i$-th entity. In this way, $g_i$ can better encode the time-series information of the corresponding entity after the model is trained by minimizing the time series forecasting error. The node embeddings $E_L^t$ outputted from the final message layer are the final outputs of the GAT network. For the first message passing layer, the input $E_0^t$ is the initialized node features, for which we use $[X^t, H^{t-1}]$, $[X^t, H^{t-1}]$, and $[X^t, r^t \odot H^{t-1}]$ in $\text{GAT}_r$, $\text{GAT}_u$, and $\text{GAT}_C$, separately. The trainable parameters of three GAT networks are shared among all time steps.

**Model Training.** The ETGNN model is first trained on a multivariate time-series dataset with only the time series forecasting task. In this stage, the model optimizes the minimum squared error loss function over the output time series, but no supervision is provided for the latent relational graph inference. In other words, the graph learning module of the ETGNN model is learning a graph that minimizes the time series forecasting error, but the learned graph does not necessarily match the real latent relational graph. Afterwards, as described in Section 4.1, the trained ETGNN model is edited with the feedback about latent relations from users. The detailed editing algorithm will be described in Section 4.3.

### 4.3 Fast Model Editing

Following the problem setting in Section 4.1, the trained ETGNN is edited to change its inferred latent relational graph $A$ to the graph $A'$ provided by users, and improve the time series forecasting performance on the new graph $A'$. The editing is done by optimizing the model parameters with gradient descent to minimize a loss function $\mathcal{L}$, which is a combination of minimum squared error loss $\mathcal{L}_{\text{MSE}}$ on the outputted time series and cross entropy loss $\mathcal{L}_{\text{CE}}$ on the inferred latent relational graph as

$$\mathcal{L} = \mathcal{L}_{\text{MSE}} + \lambda \mathcal{L}_{\text{CE}}. \tag{4}$$

Here, $\lambda$ is a hyperparameter, $\mathcal{L}_{\text{MSE}}$ is calculated with the same process as training the ETGNN model before editing, i.e., randomly sampling input time series from the dataset, passing it into the ETGNN model, and computing the minimum squared error loss of the output time series.

As for $\mathcal{L}_{\mathrm{CE}}$, we adopt different weights for loss terms on different edges to balance the loss values on the edges that are edited or unedited, or have positive or negative latent relations. Let $D_{\mathrm{e}} = \{(i,j) : A_{ij} \neq A'_{ij}\}, D_{\mathrm{p}} = \{(i,j) : A_{ij} = A'_{ij}\}$ be the edges whose predicted latent relations has been changed and not changed by users, respectively, we believe in most real-world scenarios, users will only make changes in a small region of the overall latent relational graph, so the size of $D_{\mathrm{e}}$ will be smaller than $D_{\mathrm{p}}$. Besides, in some scenarios, the numbers of positive and negative latent relations are highly imbalanced. Hence, we adopt balanced weights for cross entropy loss terms on four different categories of edges. Specifically, let $D_{\mathrm{e}}^{+} = \{(i,j) : A_{ij} \neq A'_{ij}, A'_{ij} = 1\}, D_{\mathrm{e}}^{-} = \{(i,j) : A_{ij} \neq A'_{ij}, A'_{ij} = 0\}, D_{\mathrm{p}}^{+} = \{(i,j) : A_{ij} = A'_{ij}, A'_{ij} = 1\}, D_{\mathrm{p}}^{-} = \{(i,j) : A_{ij} = A'_{ij}, A'_{ij} = 0\}, \mathcal{L}_{\mathrm{CE}}$ is computed as

$$\mathcal{L}_{\mathrm{CE}} = \sum_{D \in \{D_{\mathrm{e}}^{+}, D_{\mathrm{e}}^{-}, D_{\mathrm{p}}^{+}, D_{\mathrm{p}}^{-}\}} \frac{1}{|D|} \sum_{(i,j) \in D} \left[ A'_{ij} \log p_{ij} + (1 - A'_{ij}) \log p_{ij} \right], \tag{5}$$

where $p_{ij} \in (0,1)$ is the probability value directly outputted from the latent graph inference module of ETGNN model.

In our experiments, we edit the trained models with Adam (Kingma & Ba, 2015) optimizer to minimize the loss function in Equation 4. With the editing-suited architecture design, we find it sufficient for the ETGNN model to converge after only 1000~2000 steps of gradient descent iterations, which is much faster than retraining a model on the new latent relational graph $A'$ from scratch. See Section 5 for more experimental details.

### 4.4 Comparison with Prior Methods

**Comparison with GTS, MTGNN and Satorras et al. (2022).** Three early proposed temporal graph neural network models, GTS (Shang et al., 2020), MTGNN (Wu et al., 2020) and Satorras et al. (2022), are similar to our proposed ETGNN model in that they all jointly learn to infer latent relational graphs and forecast multivariate time series. However, compared with ETGNN, the architectures of these models are different from ETGNN and they are not designed to facilitate model editing. In (Satorras et al., 2022), different from ETGNN, latent relational graphs are predicted as soft connections in the form of attention weights. Also, the inferred graphs do not maintain globally constant, but will dynamically change when the input time series change, thereby disabling the use of graph editing. In GTS, a globally constant latent relational graph can be inferred, but it is not inferred bu learnable parameters that are separated among different entities, such as global embedding vectors in the ETGNN model. Therefore, it is not as easy as ETGNN for GTS to change a few latent relation predictions but preserve most other predictions in the graph. In MTGNN, although global embedding vectors similar to ETGNN are used in relational graph inference, these embedding vectors are not explicitly used in time series forecasting. Hence, it would be harder for MTGNN models to significantly improve the performance of time series forecasting after being edited.

**Comparison with GRAN.** Another early study (Liao et al., 2019) proposes a graph recurrent attention network (GRAN) model, which is similar to ETGNN on the surface. However, GRAN is proposed for a completely different problem of graph generation. Also, the architecture of GRAN is very different from our ETGNN model. Fundamentally, our model is a sequence-to-sequence recurrent neural network model, and the computation of each single unit in this recurrent model is graph attention network. However, the GRAN model in Liao et al. (2019) is fundamentally a graph neural network model, not a sequence-to-sequence model. It just adopts a recurrent module to update node embeddings in message passing layers, which can be understood as using a recurrent network to update $E_{\ell-1,i}^{t}$ to $E_{\ell-1,i}^{t}$ in Equation (3). But the recurrent module is never applied in time dimension.

## 5 Experiments

In this section, we evaluate the proposed time series model editing framework on three time series datasets, including one physics simulation dataset and two traffic datasets. We conduct experiments to validate the

Table 2: Statistics of datasets, including the total length of time series, the number of entities, the lengths of input and output time series.

| Dataset | Total Length | # Entities | Input Length | Output Length |
|---|---|---|---|---|
| Spring-10 | 10,000 | 10 | 20 | 20 |
| METR-LA | 34,272 | 207 | 12 | 12 |
| PEMS-BAY | 52,116 | 325 | 12 | 12 |

effectiveness of ETGNN in quickly incorporating edits on latent relational graphs and improving time series forecasting performances. We also compare ETGNN with other temporal GNN models in terms of their model editing capacities when giving different amounts of edit data. In addition, we verify the impact of different design choices for ETGNN architecture through ablation studies.

## 5.1 Experimental Setup

**Datasets.** We evaluate on the following three multivariate time series datasets. The statistics of these datasets are summarized in Table 2.

- **Spring-10**. This is a physics-simulated time series dataset proposed by Kipf et al. (2018). To generate the data, we initialize the positions and velocities of 10 particles in a 2D plane, and randomly add springs to connect the particles. The positions and velocities of particles are changing over time due to the effects of spring forces, which are computationally simulated by the source codes[1] of Kipf et al. (2018). We simulate the positions and velocities of 10 particles for 10,000 time steps. The spring connection graph is the ground truth latent relational graph.

- **METR-LA**. This is a traffic dataset collected from loop detectors in Los Angeles (Jagadish et al., 2014) and used by many prior multivariate time series forecasting studies (Li et al., 2018; Shang et al., 2020; Wu et al., 2020). Following prior studies, we conduct experiments on the traffic time series data generated by 207 sensors from Mar 1st 2012 to Jun 30th 2012. The ground truth latent relational graph is given as a radius cutoff graph, in which nodes are associated with sensors and a latent relation of value 1 between two nodes indicates the distances between their corresponding sensors is smaller than a pre-defined radius threshold.

- **PEMS-BAY**. Similar to METR-LA, PEMS-BAY is another widely used traffic time series dataset, which is collected by the California Transportation Agencies Performance Measurement System. This dataset contains the traffic time series data of 325 sensors in the Bay Area for six months. The ground truth graph is also given and its construction process is the same as that in METR-LA dataset.

**Baselines.** Since the main focus of this work is editing the models that jointly learn latent relational graph inference and time series forecasting, we compare our proposed ETGNN model with the following two baselines.

- **GTS** (Shang et al., 2020). In this model, the latent relational graph is first predicted from the complete time series by a 1D convolutional neural network model. The graph is then passed into a diffusion convolutional recurrent neural network (DCRNN) (Li et al., 2018) model to forecast the time series. Different from ETGNN, no parameters like global embeddings are used in graph inference.

- **MTGNN** (Wu et al., 2020). This model predicts the latent relational graph from node embeddings that are similar to the global embeddings used in our ETGNN, then forecasts the time series with a temporal model composed of multiple graph convolution and temporal convolution modules. Different from ETGNN, the node embeddings are not used in time series forecasting.

---

[1] https://github.com/ethanfetaya/NRI

Table 3: The editing performance of GTS, MTGNN, and ETGNN on three datasets. Here, we present the F1 scores in latent graph inference before and after the models are edited, and the MAE in time series forecasting before and after the models are edited, and the MAE reduction percentages $\Delta$ after the models are edited. We show that after being edited, ETGNN model achieves the highest F1 score and MAE reduction percentage.

| Dataset | Method | F1 score | | MAE | | |
| --- | --- | --- | --- | --- | --- | --- |
| | | Before editing | After editing | Before editing | After editing | $\Delta$ |
| Spring-10 | GTS | 0.422 | **1.000** | 0.0918 | 0.0789 | -14.05% |
| | MTGNN | 0.799 | **1.000** | 0.0256 | 0.0284 | +10.94% |
| | ETGNN (ours) | 0.600 | **1.000** | 0.1074 | 0.0641 | **-40.32%** |
| METR-LA | GTS | 0.529 | 0.909 | 5.8565 | 5.3673 | -8.35% |
| | MTGNN | 0.077 | 0.904 | 4.6247 | 4.6921 | +1.46% |
| | ETGNN (ours) | 0.062 | **0.924** | 5.3883 | 4.8006 | **-10.91%** |
| PEMS-BAY | GTS | 0.037 | 0.892 | 2.1834 | 2.0367 | -6.72% |
| | MTGNN | 0.050 | 0.905 | 1.6281 | 1.6408 | +0.78% |
| | ETGNN (ours) | 0.106 | **0.914** | 2.3458 | 1.8090 | **-22.88%** |

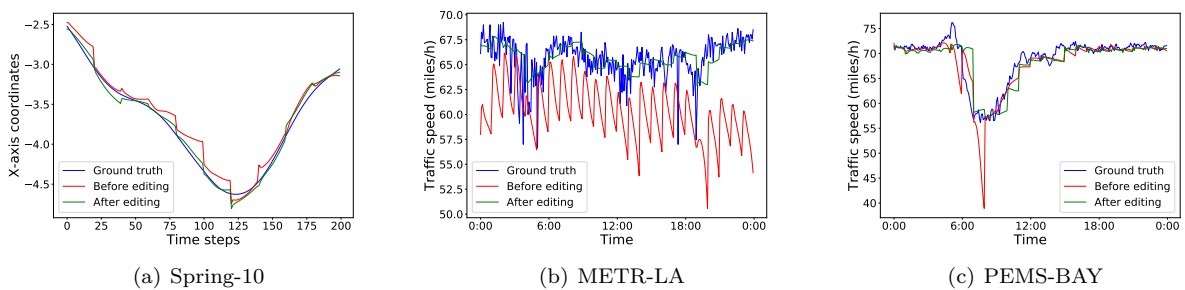

(a) Spring-10       (b) METR-LA       (c) PEMS-BAY

Figure 3: Illustration of time series forecast results on three datasets. We compare the time series forecasted by ETGNN models before and after editing with the ground truth time series. We visualize the results of a 200-step x-axis coordinate series of the first instance for Spring-10 dataset, the results of traffic series of the 31st instance on March 2nd 2012 for METR-LA dataset, and the results of traffic series of the first instance on January 2nd 2017 for PEMS-BAY dataset. We can visually observe that editing ETGNN models can reduce the gap between the forecasted and ground truth time series.

**Evaluation Pipeline.** Since the main focus of this work is editing temporal graph neural network models, we compare our ETGNN with GTS and MTGNN in terms of their improvement in graph inference and time series forecasting after being edited. Specifically, we first train models on the time series dataset, then edit the models with the "user feedback" data that are synthesized from the ground truth latent relational graph. For all models, we use the editing algorithm in Section 4.3. We set learning rate to 0.001, gradient descent step number to 2000 and $\lambda$ in Equation (4) to 5.0. We evaluate the graph inference performance using the F1 score of latent relation predictions over all pairwise nodes, and the time series forecasting performance by mean absolute error (MAE) averaged over all output time steps. To reflect the editing capacities of different models, we compare their F1 scores in latent graph inference and reduction percentages in time series forecasting MAE after editing them with the same procedure.

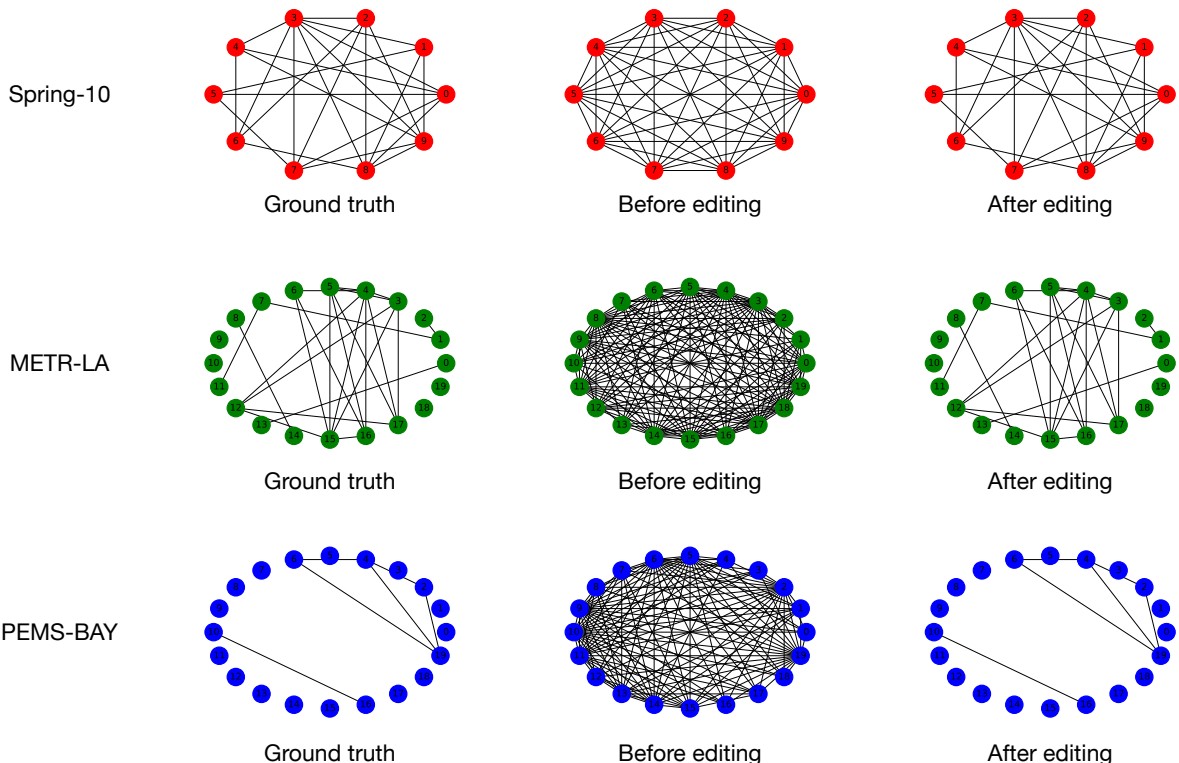

Figure 4: Illustration of latent relational graph inference results on three datasets. We compare the latent graphs inferred by our ETGNN models before and after editing to the ground truth latent graphs. Note that there are hundreds of instances in METR-LA and PEMS-BAY datasets, but we only visualize the subgraphs formed by the first 20 instances for the convenience of visualization. We can visually observe that editing ETGNN models can correct the errors in latent graph inference.

## 5.2 Experimental Results

We first provide the complete ground truth graph as the "user feedback", and edit models to incorporate the feedback information into their predicted latent relational graph and forecasted time series. Ideally, the model with strong editing capacities can effectively correct the mistakenly predicted latent relations and improve the performance of time series forecasting, after being edited by as low as 2000 gradient descent steps. For our method and baseline methods, we summarize their achieved F1 scores in latent graph inference and reduction percentages in time series forecasting MAE on three datasets in Table 3. From the results, we can clearly observe that our ETGNN model achieves the highest value in F1 score and MAE reduction percentage after being edited. Particularly, we can find that after being edited with the same procedure, though all methods can correct most mistakenly inferred latent relations and achieve high F1 score, GTS only achieves minor improvements in time series forecasting and MTGNN even fails to achieve improvements. Differently, editing ETGNN models enables them to not only accurately infer latent graphs and consistently achieve higher than 0.9 in F1 scores, but also reduce the time series forecasting error by a significant margin of 10%~40%. All these results demonstrate the powerful editing capacity of ETGNN, showing that ETGNN can make good use of the information from the user feedback data and quickly adjust its predicted latent graph and time series.

For the convenience of visually observing how the inferred latent graphs and forecasted time series change after ETGNN models are edited, we visualize the time series forecasts (Figure 3) and inferred latent graphs (Figure 4) from ETGNN models before and after editing, and compare the results to ground truth time series and graphs. These visualizations show that our editing algorithm can effectively correct the errors in latent graph inference, and reduce the gap between the forecasted and ground truth time series.

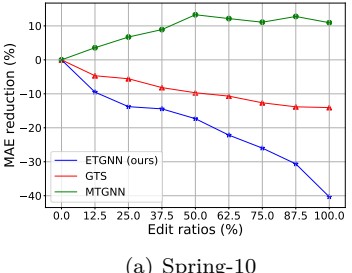 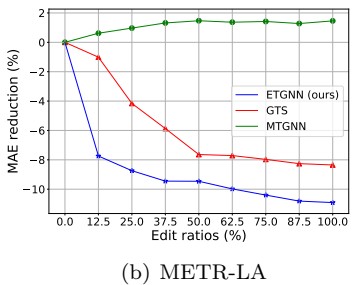 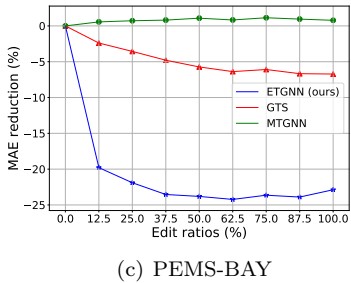

(a) Spring-10             (b) METR-LA             (c) PEMS-BAY

Figure 5: The reduction (%) in time series forecasting MAE of ETGNN and GTS methods after editing different ratios of error edges on three datasets. We show that ETGNN consistently achieves larger MAE reduction in time series forecasting on the editing data with different sizes.

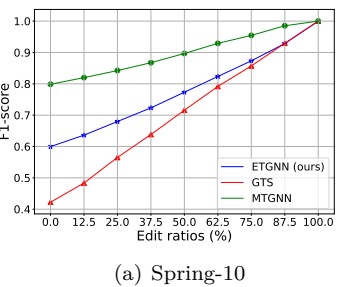 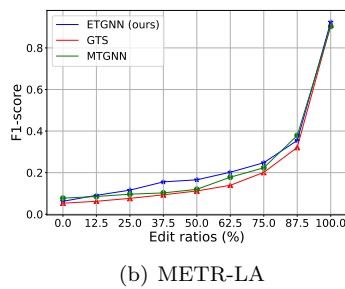 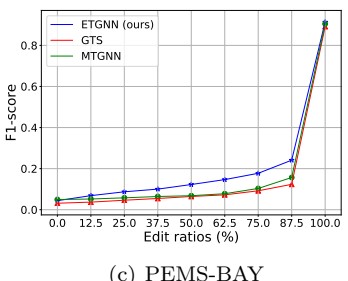

(a) Spring-10             (b) METR-LA             (c) PEMS-BAY

Figure 6: The graph inference F1-score of ETGNN and GTS methods after editing different ratios of error edges on three datasets. Three models can all be edited to correct their graph inference errors when they are given the editing data with different sizes.

To further demonstrate the advantages of our method, we compare our ETGNN to GTS and MTGNN in editing capacities with different amounts of editing data. Specifically, instead of providing the complete ground truth graph, we only correct a portion of mistakenly inferred latent relations in the graph inferred by the original model, and provide this partially corrected graph as the "user feedback" to edit the model. This setting can simulate the scenarios where real-world users do not correct all errors in the graph in one time, but try to observe the improvements from editing a portion of errors. In our experiments, we try correcting the errors in the graph with ratios from 12.5% to 100.0%, and illustrate the changing curves of MAE reductions in time series forecasting and F1 scores in latent graph inference in Figure 5 and Figure 6, respectively. From these two curves, we can find that on the editing data with different sizes, all models can be edited to effectively correct their errors in latent graph inference, but ETGNN consistently achieves larger MAE reduction in time series forecasting. Especially, on METR-LA and PEMS-BAY datasets, even given a little amount of editing data, ETGNN can still effectively make use of its information and achieves significant reduction in time series forecasting MAE, while GTS and MTGNN fail to achieve it.

### 5.3 Ablation Study

In addition to comparing ETGNN to baseline methods in their editing capacities, we conduct two ablation studies to show that (1) editing model parameters through gradient descent based editing algorithm is important to incorporate information from the user feedback, and (2) the use of global embedding vectors in both latent graph inference and time series forecasting improves the editing capacity.

Table 4: The reduction (%) in time series forecasting MAE of different ETGNN models, in which one ETGNN model is not edited but simply uses the new latent graph provided by the user to forecast time series, and the other ETGNN model is edited on the new latent graph. We show that simply using the new latent graph but skipping model editing fails to achieve significant improvements in time series forecasting.

| Method | Spring-10 | METR-LA | PEMS-BAY |
|--------|-----------|---------|----------|
| No editing | -11.73% | -0.77% | -11.25% |
| Editing | **-40.32%** | **-10.91%** | **-22.88%** |

Table 5: The editing performance of ETGNN variant and ETGNN on three datasets. Here, ETGNN variant is the ETGNN model in which global embeddings do not directly affect the time series forecast module. We show that ETGNN always achieves higher reduction in time series forecasting MAE than ETGNN variant. This demonstrates the advantages of involving global embeddings to time series forecasting.

| Dataset | Method | F1 score | | MAE | | |
|---------|--------|----------|---------|--------|---------|---|
| | | Before editing | After editing | Before editing | After editing | Δ |
| Spring-10 | ETGNN variant | 0.531 | **1.000** | 0.1062 | 0.0742 | -30.13% |
| | ETGNN | 0.600 | **1.000** | 0.1074 | 0.0641 | **-40.32%** |
| METR-LA | ETGNN variant | 0.067 | 0.920 | 5.4384 | 4.9241 | -8.35% |
| | ETGNN | 0.062 | **0.924** | 5.3883 | 4.8006 | **-9.46%** |
| PEMS-BAY | ETGNN variant | 0.043 | 0.906 | 2.3781 | 1.8546 | -22.01% |
| | ETGNN | 0.106 | **0.914** | 2.3458 | 1.8090 | **-22.88%** |

**Ablation on Editing Model Parameters.** In our proposed temporal model editing framework, given the new latent graph, the editing is done by updating model parameters so that the predicted latent graph matches the new graph and the time series forecasting performance error can be reduced on the new graph. However, one more simple way is to directly use the new latent graph to forecast time series with the time series forecasting module of ETGNN, but skip editing any model parameters. On the three benchmark datasets used in previous experiments, we try this simple strategy of taking the complete ground truth graph as the "user feedback" and compare the reduction in time series forecasting MAE with our editing algorithm. The results on three datasets are presented in Table 4. They show that simply replacing the latent graph but skipping model editing leads to significantly lower reduction in time series forecasting.

**Ablation on Global Embeddings.** As discussed in Section 4.2, a significant design of our ETGNN model is using global embeddings in both latent graph inference and time series forecasting. We verify its benefits to improving the editing capacity by comparing with a variant of our ETGNN model, in which the global embedding vectors are not used as the learnable query vector for the attention score calculation of Equation (3). Instead, another randomly initialized vector is used as the learnable query vector, which is the same as the original GATv2 model (Brody et al., 2022). In this way, global embeddings only have direct effect on the latent graph inference in the variant model. On the three benchmark datasets, we evaluate the editing performance of this ETGNN variant and the original ETGNN model using the complete ground truth graph as the "user feedback", and present the results in Table 5. The results clearly show that after being edited, though the two models achieve similar F1 scores of latent graph prediction, the original ETGNN model achieves higher reduction in time series forecasting MAE. This demonstrates the advantages of involving global embeddings to time series forecasting.

# 6 Conclusion

In this work, we study the problem of editing multivariate time series forecasting models from user feedback on the latent relations. We formulate the editing problem as editing the latent relational graph learned by the model and improving time series forecasting on the new latent graph. We propose a novel framework to solve this problem, which includes an editing-suited model, named ETGNN, and a gradient descent based editing algorithm. ETGNN first infers latent relational graph, then forecasts time series from the latent relational graph. In ETGNN, every entity is associated with a global embedding and all global embeddings are used in the prediction of both latent graphs and time series. The use of these global embeddings facilitates effectively editing the predictions of a few entities while unrelated entities are not affected. From comprehensive experiments on benchmark datasets, we show that our ETGNN has more powerful editing capacities than prior methods. We hope our exploration in this work will motivate more research on the problem of efficiently editing multivariate time series forecasting models in the future.

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
