# OpenReview forum: "Editable Temporal Graph Neural Network for Interactive Time Series Forecasting"
_TMLR — Rejected by TMLR_

### Review · Reviewer_NLMj · 2023-06-11

**Summary Of Contributions:**

This paper studies the problem of Interactive Time Series Forecasting and proposes a method named Editable Temporal Graph Neural Network (ETGNN).It uses global embeddings in both graph learning and time series forecasting.

**Audience:**

Yes

**Claims And Evidence:**

No

**Requested Changes:**

1. Add more detailed caption of Fig.1.
2. Please add more baselines.
3. Please add sensitivity analysis.
4. Please discuss and compare with graph structure learning methods.
5. Please talk about the difficulty of getting underlying graph structures in forecasting problems.

**Strengths And Weaknesses:**

Pros.

1. The problem is very important.
2. The writing is good.

Cons.
1. The caption of Fig.1 is needed in detail to make people understand your main idea.
2. The compared methods are limited (two at 2020), which makes the experiments less persuasive. Please add more baselines.
3. Lack of sensitivity analysis.
4. I think graph editing is kind of like graph structure learning, as your problem doesn't have the known dependency graph. Please compare your method with these methods.
5. I also doubt whether there is practical scenarios of this problem. For example, in traffic flow forecasting, graph structure is always accessible from the road map. Moreover, does the Figure. 4 shows the common scenarios? how is the performance of graph structural learning.

---

> ### Author Response · Authors · 2023-08-09
> **Response to Reviewer NLMj**
>
> Thank you for your feedback and comments. We hope your concerns and questions can be addressed by the following responses.
>
> > **Q1: The caption of Fig.1 is needed in detail to make people understand your main idea.**
>
> Thanks for your suggestion. We have added more details about our contribution and ideas to the caption of Figure 1 in the revision of our paper.
>
> > **Q2: The compared methods are limited (two at 2020), which makes the experiments less persuasive. Please add more baselines.**
>
> As we focus on the task of editing temporal graph neural network models through their inferred latent graphs, we only compare with models that jointly do latent relational graph inference and time series forecasting. To the best of our knowledge, there are only three models that tackles the same setting: GTS [1], MTGNN [2], and the model proposed in [3]. We have compared with GTS and MTGNN, and [3] does not release their codes so we cannot compare with it.
>
> We would appreciate any concrete pointers to the baselines that are comparable to our method.
>
> [1] Wu, Zonghan, et al. "Connecting the dots: Multivariate time series forecasting with graph neural networks." Proceedings of the 26th ACM SIGKDD international conference on knowledge discovery & data mining. 2020.
> [2] Shang, Chao, Jie Chen, and Jinbo Bi. "Discrete Graph Structure Learning for Forecasting Multiple Time Series." International Conference on Learning Representations. 2020.
> [3] Satorras, Victor Garcia, Syama Sundar Rangapuram, and Tim Januschowski. "Multivariate time series forecasting with latent graph inference." arXiv preprint arXiv:2203.03423 (2022).
>
> > **Q3: Lack of sensitivity analysis.**
>
> In Section 5.2, we conducted experiments to understand the impact on forecasting performance after editing different ratios of error edges in the latent relational graph. This is a type of sensitivity analysis of our model w.r.t the amount of edited edges.
>
> We welcome your suggestions on specific sensitivity analysis experiments.
>
> > **Q4: I think graph editing is kind of like graph structure learning, as your problem doesn't have the known dependency graph. Please compare your method with these methods.**
>
> Graph editing is **NOT** graph structure learning. We focus on learning graph structures and multivariate time series forecasting jointly, which is fundamentally different from graph structure learning where the goal is to only learn the graph structure from data. They are not directly comparable because the model outputs are very different.
>
> > **Q5: I also doubt whether there is practical scenarios of this problem. For example, in traffic flow forecasting, graph structure is always accessible from the road map. Moreover, does the Figure. 4 shows the common scenarios? how is the performance of graph structural learning.**
>
> - There exist many practical scenarios of our problem as accurate graph structures are not always available. We provided many motivating scenarios in the Introduction section. Here we give more details.
>
> - For instance, in the problem of predicting prices of goods, obtaining the latent relational graph of goods may be very helpful because the prices of certain goods are mutually affected and may have the same changing trends in a time (such as cellphones and chips). However, real-world application scenarios may require to predict the prices of several thousands of goods, and producing an accurate latent relational graph for so many goods is very hard and laborious. For such large-scale problems, we believe the accurate graph structure is often not accessible, and learning latent relational graphs from time series by temporal models and further editing models by user feedback on latent relational graphs is a more practical solution.
>
> - We use traffic datasets only as benchmark datasets to demonstrate the effectiveness of our proposed method. Figure 4 visualizes graphs learned on benchmark datasets, but the graphs in common real-world large-scale problems may be more complicated. The graph structural learning performance is evaluated by F1 scores of latent graph prediction, which are presented in Table 3. Results show that our ETGNN model can achieve close-to-perfect performance (with the F1-score of larger than 0.9) for latent graph prediction after editing.

---

### Review · Reviewer_KV8K · 2023-06-13

**Summary Of Contributions:**

In this paper, the authors deal with the problem of time series forecasting. In cases where multiple dependent variables are present in the time series data, capturing the relationships between them can improve the forecasting performance. Previous approaches have proposed the construction of a graph where nodes represent the different variables and where some approach is employed to learn the graph structure (i.e., edges). This paper proposes an editing algorithm that allows the model to incorporate user feedback and make edits on the learned graphs before obtaining forecasts. The proposed approach is experimentally evaluated on some datasets and the results show that it leads to significant improvements in time series forecasting.

**Audience:**

Yes

**Broader Impact Concerns:**

--

**Claims And Evidence:**

Yes

**Requested Changes:**

- The authors need to better motivate the proposed approach, provide examples of real-world problems to which the proposed method could be applied and design a realistic experimental setup.
- Please, explain what are the limitations of the proposed editing approach and whether a more sophisticated approach could lead to further improvements.
- Stronger empirical results compared to the baselines (e.g., MTGNN)
- Clearer writing separating the actual contribution from prior work (e.g., temporal graph neural networks that can jointly infer the relational graph structure and forecast time series).
- Explain why model editing does not always yield improvements. For example, model editing offers no performance gains to MTGNN.

**Strengths And Weaknesses:**

Strengths:

- The paper is easy to read and the proposed model editing approach is clearly presented.

- Even though the proposed model editing approach is simple, it leads the ETGNN model to significant performance improvements.

Weaknesses:

- In my opinion, the proposed approach is not properly motivated. First of all, there is no direct analogy with the use of model editing in NLP applications where the user typically has a continuous interaction with the large language model and can easily (in most cases) correct the errors the model makes. Furthermore, if the graph structure is known, why one would need to first learn it and then edit it? Is the employed experimental setup realistic?

- The methodological contribution of the paper is minimal with respect to other works such as GTS. The main contribution is presented in subsection 4.3 and it only consists of an extra term that is added to the loss function and which compares the learned edges against those provided by the users.

- The significance of the reported results is unclear. In Table 3, we can see that the best performing model in the task of forecasting is MTGNN with no editing. This model outperforms the proposed ETGNN model. The results reported in this Table also question the usefulness of model editing in this setting since no editing has been performed in the case of the best performing method.

---

> ### Author Response · Authors · 2023-08-09
> **Response to Reviewer KV8K Part 1**
>
> Thank you for your constructive feedback and comments. We hope your concerns and questions can be addressed by the following responses.
>
> > **Q1: In my opinion, the proposed approach is not properly motivated. First of all, there is no direct analogy with the use of model editing in NLP applications where the user typically has a continuous interaction with the large language model and can easily (in most cases) correct the errors the model makes. Furthermore, if the graph structure is known, why one would need to first learn it and then edit it? Is the employed experimental setup realistic? The authors need to better motivate the proposed approach, provide examples of real-world problems to which the proposed method could be applied and design a realistic experimental setup.**
>
> - First, model editing has been studied intensitively in NLP applications [1,2]
>
> - Second, we focus on scenarios where the graph is unknown and need to be inferred. Even when the graph structure is known, it may not be perfectly accurate and hence needs user editing.
>
> - Our experiments simulate this scenario. We use the ground truth graph to simulate the feedback from the user and evaluate the accuracy of models in graph prediction. Particularly, in Section 5.2, we evaluate the performance of our forecasting model after correcting a small portion of the mistakenly inferred graph, which imitates the real user's behavior of only editing latent relations. Hence, we believe that our experimental setup is realistic and the promising results of our method demonstrate its potentials in practical use.
>
> [1] Mitchell, E., Lin, C., Bosselut, A., Manning, C. D., & Finn, C. (2022, June). Memory-based model editing at scale. In International Conference on Machine Learning (pp. 15817-15831). PMLR.
>
> [2] Meng, K., Sharma, A. S., Andonian, A. J., Belinkov, Y., & Bau, D. (2022, September). Mass-Editing Memory in a Transformer. In The Eleventh International Conference on Learning Representations.
>
> > **Q2: The methodological contribution of the paper is minimal with respect to other works such as GTS. The main contribution is presented in subsection 4.3 and it only consists of an extra term that is added to the loss function and which compares the learned edges against those provided by the users.**
>
> - We respectfully disagree. Our contribution includes (1) introducing the new problem setting to the temporal graph learning community (2) designing a novel architecture of ETGNN model to facilitate model editing and (3) designing experiments to validate our hypothesis in the new setting.
>
> - Specifically for our proposed ETGNN model, we assign one global embedding vector to every entity, and make each global embedding vector only have large impacts on both the latent relation inference and time series forecasting of its corresponding entity. This novel design is important to faclitate model editing (see the third paragraph in Section 4.2 for details). We have experimentally shown that ETGNN has more powerful model editing capacities than other temporal graph neural networks such as GTS in Section 5.2 and the importance of this design in the ablation studies of Section 5.3.

---

> ### Author Response · Authors · 2023-08-09
> **Response to Reviewer KV8K Part 2**
>
> > **Q3: The significance of the reported results is unclear. In Table 3, we can see that the best performing model in the task of forecasting is MTGNN with no editing. This model outperforms the proposed ETGNN model. The results reported in this Table also question the usefulness of model editing in this setting since no editing has been performed in the case of the best performing method. Explain why model editing does not always yield improvements. For example, model editing offers no performance gains to MTGNN.**
>
> - MTGNN has strong time series forecasting performance, but such performance does not come from the latent relational graph.  In fact, we tried removing the graph inference module of MTGNN and training MTGNN with randomly graphs. Surprisingly, We found that MTGNN trained on random graphs has similar performance to the one trained on the ground truth graph.
>
> - This shows that the usefulness of model editing is dependent on the model as well. Some models such as as MTGNN does not benefit from editing. Our proposed model and the GTS model we compared are more suitable for the model editing task.
>
>
> > **Q4: Please, explain what are the limitations of the proposed editing approach and whether a more sophisticated approach could lead to further improvements.**
>
> The proposed editing approach needs around 1000~2000 steps of gradient descent to converge. Though it is faster than retraining the model from scratch, we think there still exists space in improving the editing speed. We think some approaches can be used to accelerate the convergence, e.g., using higher learning rates for global embeddings when editing the model.
>
> > **Q5: Clearer writing separating the actual contribution from prior work (e.g., temporal graph neural networks that can jointly infer the relational graph structure and forecast time series).**
>
> Thanks for your suggestion. We have already clarified the difference between our methods and three previous methods GTS [1], MTGNN [2], and [3] in the first paragraph of Section 4.4.
>
> [1] Wu, Zonghan, et al. "Connecting the dots: Multivariate time series forecasting with graph neural networks." Proceedings of the 26th ACM SIGKDD international conference on knowledge discovery & data mining. 2020.
> [2] Shang, Chao, Jie Chen, and Jinbo Bi. "Discrete Graph Structure Learning for Forecasting Multiple Time Series." International Conference on Learning Representations. 2020.
> [3] Satorras, Victor Garcia, Syama Sundar Rangapuram, and Tim Januschowski. "Multivariate time series forecasting with latent graph inference." arXiv preprint arXiv:2203.03423 (2022).

---

### Review · Reviewer_EobN · 2023-07-26

**Summary Of Contributions:**

This paper considers the problem of multivariate time series forecasting when several (dependent) time series need to be predicted. The proposed approach simultaneously makes predictions and learns the edges of a hidden graph. The main challenge addressed by the authors is how to allow users to easily modify incorrectly predicted edges when information about some relations is available. For this, global embeddings are used: they affect both time series and graph structure predictions. The experiments show that such edits improve both the quality of graph reconstruction and time series forecasting.

**Audience:**

Yes

**Claims And Evidence:**

Yes

**Requested Changes:**

My main concerns are listed in “Weaknesses” above.

Some other non-critical suggestions are listed below:

- In Section 4.4, the authors compare the proposed approach with existing methods. A similar discussion about Satorras et al. (2022) would also be helpful.
- Some model/training details are not specified. For instance, how the parameter lambda is chosen?
- In the experiments, the complete actual graph is used as supervision, and it can be changed by the editing algorithm. What would happen if the graph is used as is and is not changed (but the time series prediction model is trained)? As I understand, in Section 5.3, a different comparison is performed.

The paper contains some typos and grammatical errors; here are some examples:
- “to improve the accuracy and explainable of the model”
- “at which, different sensors” (misplaced comma)
- “the parameter updates are use”
- “many works propose to jointly learning”
- “we are the first one”
- “that is particular well-suited”
- “with other temporal GNN model”
- Incorrectly formatted quotation marks in some cases (e.g., the beginning of Section 5.2)
- “reduce the the time”
- “this work will motivates”

**Strengths And Weaknesses:**

Strengths:
- The paper is well-written and easy to follow.
- The proposed solution is reasonable, and the experiments confirm that graph modifications improve the quality of graph reconstruction and time series forecasting.

Weaknesses:
- According to the results in Table 3, the best time-series predictions are achieved by MTGNN before editing. And the quality of this model does not improve with changing the graph. Thus, it can be possible that MTGNN learns a graph that is more suitable for prediction. So, my main concern is the motivation for editing – it can be the case that models, while not being able to learn the actual graph, can learn other structures that are better for prediction.
- Also, in the experiments (Section 5.1), the complete true graph is used as a supervision. Thus, it is not surprising that the graph can be recovered with good precision. The setup with a smaller fraction of corrected edges (as considered in Section 5.2) is more practical, and here the performance of graph recovery is significantly worse.
- As can be seen from Figure 4, the proposed model tends to predict an almost complete graph that is very far from the true one. More discussions about the quality of graph reconstruction before editing would be helpful.

---

> ### Author Response · Authors · 2023-08-09
> **Response to Reviewer EobN**
>
> Thank you for your valuable feedback and comments. We hope your concerns and questions can be addressed by the following responses.
>
> > **Q1: According to the results in Table 3, the best time-series predictions are achieved by MTGNN before editing. And the quality of this model does not improve with changing the graph. Thus, it can be possible that MTGNN learns a graph that is more suitable for prediction. So, my main concern is the motivation for editing – it can be the case that models, while not being able to learn the actual graph, can learn other structures that are better for prediction.**
>
> MTGNN does not learn a graph that is more suitable for prediction. Actually, the reason why its time series forecasting performance does not improve with changing the graph is because its strong forecasting performance does not come from the latent relational graph.  In fact, we tried removing the graph inference module of MTGNN and training MTGNN with randomly graphs. Surprisingly, We found that MTGNN trained on random graphs has similar performance to the one trained on the ground truth graph.
>
> > **Q2: Also, in the experiments (Section 5.1), the complete true graph is used as a supervision. Thus, it is not surprising that the graph can be recovered with good precision. The setup with a smaller fraction of corrected edges (as considered in Section 5.2) is more practical, and here the performance of graph recovery is significantly worse.**
>
> Our central target is to improve the performance of time series forecasting by editing latent relational graphs, not to recover the exact ground truth graphs. When providing a smaller fraction of corrected edges, it is not strange that the model cannot recover the ground truth graph as it may only correct edges in the provided editing data. However, Figure 5 shows that the decrease in time series forecasting MAE of the ETGNN model is large and much larger than baseline methods. This shows that the editing capacity of ETGNN model is good, and will achieve more improvements in time series forecasting when a small number of corrected edges are provided.
>
> > **Q3: As can be seen from Figure 4, the proposed model tends to predict an almost complete graph that is very far from the true one. More discussions about the quality of graph reconstruction before editing would be helpful.**
>
> We agree that our ETGNN model tends to predict a dense graph which may be very different from the ground truth graph. However, our central target is to improve the performance of time series forecasting by editing latent relational graphs. Hence, we focus on the reduction of MAE in time series forecasting after the latent relational graph is edited.
>
> > **Q4: In Section 4.4, the authors compare the proposed approach with existing methods. A similar discussion about Satorras et al. (2022) would also be helpful.**
>
> Thank you for the suggestion. We have added discussion about Satorras et al. (2022) to the revision of our paper.
>
> > **Q5: Some model/training details are not specified. For instance, how the parameter lambda is chosen?**
>
> $\lambda$ in Equation (4) is set to 5.0.
>
> > **Q6: In the experiments, the complete actual graph is used as supervision, and it can be changed by the editing algorithm. What would happen if the graph is used as is and is not changed (but the time series prediction model is trained)? As I understand, in Section 5.3, a different comparison is performed.**
>
> If the graph is not changed but the time series prediction model is trained, the forecasting performance MAE would be very close to the "MAE before editing" in Table 3. As models are always trained to converge before editing, continuing to train them by minimizing time series forecasting error but not editing latent relational graphs does not improve the performance.
>
> > **Q7: The paper contains some typos and grammatical errors.**
>
> We have proofread our paper and corrected typos or grammatical errors in the revision of our paper.

---

> > ### Comment · Reviewer_EobN · 2023-08-16
> >
> > Thank you for the response! However, some of my concerns remain: on the one hand, it is desirable to be able to use graph supervision (and the accuracy of graph reconstruction is measured). On the other hand, as written in the response, the "central target is to improve the performance of time series forecasting by editing latent relational graphs, not to recover the exact ground truth graphs". However, better performance can be achieved without using any graph (MTGNN before editing). Thus, whether graph supervision is helpful for the task in general is not clear.
> >
> > Small clarification about Q6 (to correct a possible misunderstanding here): here I mean that the model uses the actual (ground truth) graph as is, not modifying it during the training.

---

### Decision · Action_Editors · 2023-08-26

**Recommendation:** Reject

**Comment:**

The reviewers unanimously recommend rejection and I support their recommendation mostly because of weak empirical results. In more detail, Table 3 reports that the best performance is achieved by the MTGNN model that does not exploit editing. Therefore, it is not clear if modifying a graph is needed in applications, which makes the motivation of the main idea highly questionable. I recommend the authors to add experimental results that would highlight the practical benefits of using editing.

**Audience:**

Yes

**Claims And Evidence:**

As pointed out by the reviewer EobN, the empirical results do not provide sufficient evidence for the necessity of graph editing (see Table 3).

**Resubmission Of Major Revision:**

The authors may consider submitting a major revision at a later time.